# Qualitative Verbal Fluency Components as Prognostic Factors for Developing Alzheimer’s Dementia and Mild Cognitive Impairment: Results from the Population-Based HELIAD Cohort

**DOI:** 10.3390/medicina58121814

**Published:** 2022-12-09

**Authors:** Ioannis Liampas, Vasiliki Folia, Elli Zoupa, Vasileios Siokas, Mary Yannakoulia, Paraskevi Sakka, Georgios Hadjigeorgiou, Nikolaos Scarmeas, Efthimios Dardiotis, Mary H. Kosmidis

**Affiliations:** 1Department of Neurology, University Hospital of Larissa, School of Medicine, University of Thessaly, Mezourlo Hill, 41110 Larissa, Greece; 2Lab of Cognitive Neuroscience, School of Psychology, Aristotle University of Thessaloniki, University Campus, 54124 Thessaloniki, Greece; 3Department of Nutrition and Dietetics, Harokopio University, 70 El. Venizelou, 17671 Athens, Greece; 4Athens Alzheimer’s Association, 89 M. Mousourou & 33 Stilponos St, 11636 Athens, Greece; 5School of Medicine, University of Cyprus, 93 Agiou Nikolaou St, Engomi, Nicosia 2408, Cyprus; 61st Department of Neurology, Aiginition Hospital, National and Kapodistrian University of Athens Medical School, 72-74 Vassilissis Sofias Ave, 11528 Athens, Greece; 7Taub Institute for Research in Alzheimer’s Disease and the Aging Brain, The Gertrude H. Sergievsky Center, Department of Neurology, Columbia University, 710 West 168th St, New York, NY 10032, USA

**Keywords:** verbal fluency, clusters, switches, intrusions, perseverations

## Abstract

*Background and Objectives*: The aim of the present study was to investigate the prognostic value of the qualitative components of verbal fluency (clustering, switching, intrusions, and perseverations) on the development of mild cognitive impairment (MCI) and dementia. *Materials and Methods*: Participants were drawn from the multidisciplinary, population-based, prospective HELIAD (Hellenic Longitudinal Investigation of Aging and Diet) cohort. Two participant sets were separately analysed: those with normal cognition and MCI at baseline. Verbal fluency was assessed via one category and one letter fluency task. Separate Cox proportional hazards regressions adjusted for important sociodemographic parameters were performed for each qualitative semantic and phonemic verbal fluency component. *Results*: There were 955 cognitively normal (CN), older (72.9 years ±4.9), predominantly female (~60%) individuals with available follow-up assessments after a mean of 3.09 years (±0.83). Among them, 34 developed dementia at follow-up (29 of whom progressed to Alzheimer’s dementia (AD)), 160 developed MCI, and 761 remained CN. Each additional perseveration on the semantic condition increased the risk of developing all-cause dementia and AD by 52% and 55%, respectively. Of note, participants with two or more perseverations on the semantic task presented a much more prominent risk for incident dementia compared to those with one or no perseverations. Among the remaining qualitative indices, none were associated with the hazard of developing all-cause dementia, AD, and MCI at follow-up. *Conclusions*: Perseverations on the semantic fluency condition were related to an increased risk of incident all-cause dementia or AD in older, CN individuals.

## 1. Introduction

Spontaneous language production and semantic memory can be concomitantly assessed via verbal fluency (VF), which typically comprises a semantic (SVF) and a phonemic (PVF) task [1,2]. VF is widely used during neuropsychological evaluations, owing to its simple and quick administration and sensitivity in revealing cognitive dysfunction at early clinical stages [3,4]. Both VF conditions (SVF and PVF) rely on frontal lobe operations (for the initiation, organization, and monitoring of word production), while SVF additionally demands a constrained search of exemplars from a superordinate category [5,6]. Therefore, SVF is additionally heavily dependent on semantic memory/temporal lobe-based processes [5,6]. PVF performance is instead accomplished through a less constrained lexical search from a broader set of vocabular exemplars [5,6]. VF scoring typically refers to the total word production on each condition [7,8]. Word production in the SVF condition has been suggested to be of crucial clinical importance in the preclinical stages of cognitive dysfunction. Specifically, it has been consistently associated with the risk of incident mild cognitive impairment (MCI) and dementia, especially of the Alzheimer’s disease (AD) type [9,10]. These associations are mainly attributed to the potential presence of premorbid neurodegenerative alterations, prior to the formal identification of dementia or MCI [9,10].

Apart from word production, however, useful prognostic information may also arise from the qualitative assessment of VF tasks, i.e., clustering and switching techniques, numbers of intrusions, and perseverations. These indices have even been proposed to be more sensitive than total word production in the early detection and differential diagnosis of cognitive impairment [11,12,13,14]. Clustering refers to the production of related words within a given semantic or phonemic subcategory, while switching refers to the efficient shifting between clusters [13]. The former relies heavily on temporal-lobe-based processes (semantic storage), while the latter depends mainly on frontal lobe operations (executive function—cognitive flexibility) [13]. Clustering is considered a relatively unconscious and automatic process, particularly vital in SVF [14,15]. Switching, on the other hand, is a more deliberate operation based on conscious decisions to shift from one subcategory to another and is equally useful in SVF and PVF [14,15]. In addition to clustering and switching, errors constitute another important qualitative component of VF. Errors are typically divided into two subcategories: out-of-category words (intrusions) and repeated words (perseverations). It has been proposed that recurrent perseverations and intrusions are caused by frontal dysfunction and particularly by deficits in working memory, attention, and inhibition [16]. Even though their use is limited in clinical practice, previous research has occasionally reported some utility in the assessment of several neurological conditions including dementia [17,18,19,20].

To date, there are only a few studies that have indicated a potential role for qualitative VF indices in the preclinical stages of dementia or cognitive impairment. Specifically, previous research has associated perseverations on the SVF condition with an increased risk of incident MCI or dementia [19]. Of note, semantic perseverations have been found to increase steeply during the last two years prior to the onset of AD [11]. Among the remaining qualitative variables, only the mean semantic cluster size has been reported to possess predictive properties regarding the development of AD among participants with memory complaints (its predictive ability was comparable to the total word production on the SVF) [21]. Consequently, the aim of the present study was to investigate the prognostic value of clustering and switching techniques and number of intrusions and perseverations in the preclinical stages of MCI and dementia. For this purpose, we capitalized on a large data set from the population-based HELIAD (Hellenic Longitudinal Investigation of Aging and Diet) cohort [22]. The prognostic value of the aforementioned qualitative variables was explored after adjusting for global cognitive status (to account for the confounding of concomitant cognitive dysfunctions) and for important sociodemographic parameters that might confound language impairment and progression to MCI-dementia [23,24].

## 2. Materials and Methods

The present study conformed with the STROBE reporting guidelines (Strengthening the Reporting of Observational Studies in Epidemiology) [25]. Participants were drawn from the population-based HELIAD (Hellenic Longitudinal Investigation of Aging and Diet) cohort. The rationale, objectives, and key elements of the HELIAD study have been reported previously in detail [26,27,28]. In short, HELIAD is a multidisciplinary, population-based, prospective cohort primarily exploring the epidemiology of dementia, cognitive impairment, and other neuropsychiatric entities in the aging Greek population. It comprises a generational cohort including both urban and rural dwellers who grew up during World War II and the Greek Civil War, which disrupted normal operations of many services including schools for extended periods of time, leaving children with intermittent educational experiences, at best. The majority of these individuals were poorly educated in terms of formal schooling, having only received elementary school education. The Institutional Ethics Review Boards of the University of Thessaly and the Kapodistrian University of Athens approved all procedures prior to the initiation of the study. Informed consent was acquired from all participants prior to participation.

Participants were selected through random sampling from the older rosters (≥65 years) of two Greek municipalities: Marousi (suburb of Athens) and Larissa (an urban–rural area in the province of Thessaly). Extensive baseline and follow-up evaluations (~2 to 2.5 h long sessions) were carried out at approximately 3-year intervals. For the present study, participants with no dementia at baseline and available follow-up investigations were considered for potential eligibility. Collaborative assessments designated by a consortium of expert neurologists and neuropsychologists were performed at both visits; a description of the evaluations pertinent to the present article is provided below.

### 2.1. Neuropsychological Assessments and Diagnostic Procedures

A comprehensive neuropsychological assessment was carried out by trained neuropsychologists [29]: non-verbal and verbal memory (Medical College of Georgia—MCG—Complex Figure Test; Greek Verbal Learning Test), language (semantic and phonological verbal fluency; subtests of the Greek version of the Boston Diagnostic Aphasia Examination short form, namely, the Boston Naming Test short form, and selected items from the Complex Ideational Material Subtest, to assess verbal comprehension and repetition of words and phrases), visuospatial ability (Judgment of Line Orientation abbreviated form; MCG Complex Figure Test copy condition; Clock Drawing Test), attention-processing speed (Trail Making Test—TMT—Part A), and executive functioning (TMT—Part B; Verbal Fluency; Anomalous Sentence Repetition; Graphical Sequence Test; Motor Programming; months forwards and backwards) were specifically evaluated. A brief screening of global cognition and orientation was performed using the Mini-Mental State Examination (MMSE). All of the aforementioned tests were capitalized on in the diagnostic classification of the participants.

As part of a comprehensive neuropsychological assessment, verbal fluency was evaluated using a category (SVF) and a letter (PVF) fluency task [7,30]. Participants were instructed to immediately begin generating items, following the announcement of the category (objects) or letter ((α alpha), and each trial lasted for 60 s. The SVF condition was administered prior to PVF. Regarding word search and production, no instructions were given, to ensure that any cognitive strategies would be spontaneously employed by the examinees. Participants were specifically told to abstain from reporting proper nouns (on the phonemic test) as well as repetitions and word variations. The number of clusters, switches, intrusions, and perseverations for SVF and PVF were calculated separately. Any identical words or variations of a previously given word were considered as perseverations. All the words that were irrelevant to each given semantic category or letter were considered as intrusions. Finally, the total numbers of clusters and switches were calculated according to the previously reported scoring guidelines: three or more consecutive words belonging to the same semantic subcategory or two consecutive words with a strong association in the Greek language (strong pairs of words) were defined as semantic clusters; semantic switches were estimated by subtracting the total number of related words (all words belonging to a semantic cluster) from the total word production and adding that to the number of semantic clusters; three or more consecutive words beginning with the same two letters and having the same sound, or two consecutive words that differed only in a vowel sound, or words that were homophones were defined as phonemic clusters; phonemic switches were calculated by subtracting the total number of related words (all words belonging to a phonemic cluster) from the total phonemic word production and adding that to the number of phonemic clusters [30,31]. Successive words stemming from the same root (such as act–action–acting) were considered as repetitions. On the other hand, three or more words sharing a part/suffix but having a different meaning (e.g., superman–supermarket–supercilious), were considered as a cluster.

The diagnostic categorization of the participants was performed during expert consensus meetings, involving senior neurologists (E.D., G.M.H., P.S., and N.S) and a neuropsychologist (M.H.K.) [32,33]. In brief, particular focus was placed on identifying potential comorbidities that could affect cognitive performance through screening participants for medical problems and illnesses, current medications, hospitalizations, depression, anxiety, essential tremor, behavioural symptoms, neuropsychiatric symptoms, functional status, Parkinson disease (PD), dementia with Lewy bodies (DLB), and personal history of cerebrovascular disease accounting for the onset (abrupt, gradual, or insidious onset) or deterioration (stepwise deterioration, stable condition, continuous decline, or fluctuating cognition) of cognitive decline. The diagnoses of dementia and possible Alzheimer’s disease (AD) were based on the *Diagnostic and Statistical Manual of Mental Disorders-IV-Text Revision* criteria [34] and the National Institute of Neurological and Communicative Disorders and Stroke/Alzheimer Disease and Related Disorders Association criteria, respectively [35]. Mild cognitive impairment (MCI) and its subtypes were diagnosed according to the Petersen criteria [36].

### 2.2. Statistical Analysis and Outcome Measures

All statistical analyses were performed using the IBM SPSS Statistics Software Version 25 (Chicago, IL, USA). Adjusted Cox proportional hazards regressions were conducted to investigate the associations of qualitative VF components with the development of dementia and MCI at follow-up. All analyses were adjusted for the following scale variables: age at baseline (in years), years of education, and standardized MMSE scores (i.e., MMSE scores of illiterate participants were converted to the literate MMSE scale of 30). All analyses were additionally adjusted for the following categorical parameters: sex, main occupation (manual or mental), and socioeconomic status (low or high) [37]. Inclusion of the MMSE score aimed towards controlling for global cognitive function. In other words, we explored the value of qualitative VF variables over and above the general cognitive status of the participants. Time to second visit was employed as the time-to-event variable. In the case of not documenting the event of interest, participants were censored at the second visit.

First, all CN participants at baseline were analysed. Eight consecutive adjusted Cox proportional hazards regressions were performed, involving the eight qualitative VF indices as potential predictors and dementia development at follow-up as the dichotomous outcome (intercorrelations prevented the insertion of all strategy variables in a single model). Owing to the multiple comparisons, the significance threshold was corrected to α = 0.006. Exploratory analyses with AD development at follow-up as the dichotomous outcome were subsequently conducted (using the corrected criterion α = 0.006).

Second, all CN participants at baseline without a dementia diagnosis at follow-up were analysed (i.e., those who either remained CN or developed MCI at follow-up). Eight consecutive adjusted Cox proportional hazards regressions were again performed, involving the eight qualitative VF variables as potential predictors and MCI development at follow-up as the dichotomous outcome. It was not feasible to classify individuals with a follow-up diagnosis of dementia according to the outcome of interest (MCI), in view of the lack of information regarding their transitional conversion to MCI. Therefore, individuals with diagnosis of dementia at follow-up were excluded from our analyses. Again, the corrected α = 0.006 significance threshold was implemented.

Finally, a secondary analysis involving the smaller participant set of MCI individuals at baseline was performed. Eight consecutive adjusted Cox proportional hazards regressions were conducted, involving the eight qualitative VF variables as potential predictors and dementia development at follow-up as the dichotomous outcome (again, using the corrected criterion α = 0.006). Exploratory analyses with AD development at follow-up as the dichotomous outcome were subsequently conducted (according to the corrected criterion α = 0.006).

## 3. Results

### 3.1. Baseline Characteristics and Missing Data

Among the 1984 participants of the HELIAD cohort, there were 103 individuals with dementia (79 with AD, 9 with vascular dementia, 8 with dementia with LBD or PD dementia, and 7 with less common entities), 243 with MCI, 1607 who were CN, and 4 with missing data leading to inconclusive cognitive diagnosis. A subgroup of 959 CN, older (mean age = 72.9 years ±4.9), predominantly female (~60%) individuals had completed their follow-up assessments after an average of 3.09 years (±0.83). Among them, 34 had developed dementia at follow-up (29 of whom developed AD), 160 developed MCI, 761 remained CN and four 4 were missing a follow-up cognitive diagnosis (these 4 were excluded from the analysis).

Those who had developed dementia or MCI by the second visit were older at baseline, less educated, and had lower MMSE scores compared to those who remained CN (Table 1 and Table 2). In addition, a greater portion of those who were diagnosed with MCI at first visit were of low socioeconomic status and had a manual occupation compared to those who remained CN at follow-up. Regarding the qualitative VF components, CN participants who had developed dementia by the 2nd visit recorded fewer clusters and switches on the SVF, as well as fewer switches on the PVF condition. On the other hand, CN participants with a follow-up MCI diagnosis documented fewer switches on both conditions as well as more intrusions on the PVF condition.

Regarding those with MCI at baseline (N = 243), 118 older (mean age = 75.4 years ±5.0), predominantly female (~60%) participants had completed their follow-up assessments after a mean of 2.92 years (±0.89). A total of 29 individuals with MCI had developed dementia by the second visit, 25 of whom were diagnosed with AD. Those with dementia at the second visit were older, often of low socioeconomic status, had performed more poorly in MMSE, and had generated fewer related words and clusters in the SVF condition (data not shown).

### 3.2. Associations between Qualitative Verbal Fluency Indices and Incident Dementia or Mild Cognitive Impairment

Table 3 summarizes the associations of the qualitative VF components with incident all-cause dementia or AD in CN participants. Each additional perseveration in the SVF condition increased the risk of developing dementia and AD by 52% and 55%, respectively. Of note, participants with two or more perseverations at baseline presented a much more prominent risk of incident dementia compared to those who documented one or no repetitions (Figure 1). Among the remaining qualitative variables (on both SVF and PVF), none were found to be associated with the hazard of developing all-cause dementia (or AD) at follow-up.

Table 4 summarizes the associations of the qualitative VF parameters with incident all-cause dementia or AD in MCI participants. Neither SVF nor PVF indices were related to the hazard of future dementia or AD. Similarly, no qualitative parameter was linked to the risk of CN individuals progressing to MCI (Table 5).

## 4. Discussion

The present study demonstrated that perseverations on the SVF condition are related to an increased risk of incident all-cause dementia or AD in older, CN individuals. These results are in line with the findings of Pakhomov et al., who reported that perseverations on the SVF condition constitute a harbinger of MCI or dementia [19]. No other variable (i.e., numbers of clusters, switches, or intrusions) was associated with the hazard of developing dementia or AD in participants with CN. Moreover, the numbers of clusters, switches, intrusions, or perseverations were not related to a differential risk of progressing to MCI or converting from MCI to dementia or AD.

There are a few studies that have explored the neuropsychological construct of perseverations in AD-related neurodegeneration and highlighted the pivotal implication of working memory and executive control deficits. These conclusions have been mostly deducted from the temporal pattern of perseverations’ occurrence: specifically, the distance/lag between the first occurrence of an exemplar and its later repetition [17,18,38]. It has been proposed that short lags between the first occurrence of a word and its reappearance are common in aphasia and could stem from impairments in lexical selection, i.e., language dysfunction [39]. On the other hand, long intervals are more frequent in patients with AD and could be attributed to an impediment of temporary information holding and self-monitoring, i.e., working memory deficits [17,18,19,39].

Considering that neuropathological alterations tend to precede the clinical onset of AD (or other neurodegenerative entities) for years, perseverations might reflect the clinical equivalent of these early pathological processes in non-demented individuals [40]. Working memory and inhibitory control deficits have been reported to manifest in the preclinical course towards the development of AD, typically within the last 3 years before its onset [41,42,43]. Intriguingly, Auriacombe et al. have specifically investigated the preclinical course of repetitions and intrusions and reported that the rate of repetitions on the SVF task increases steeply shortly (about 2 years) before the diagnosis of AD [11]. Considering that episodic memory, language, and visuo-perceptual impairment tend to manifest as early as 4–8 years prior to the formal identification of AD [44,45,46], the detection of perseverations could represent a pivotal turning point in the preclinical neuropathological courses of older adults towards the development of AD (a prominent implication of frontal operations), ultimately leading to full-blown dementia of the AD type [47].

Although perseverations may constitute a useful preclinical marker of imminent dementia, this marker probably lacks specificity. Working memory impairment is a common feature in the course of many common neurodegenerative entities and, therefore, cannot reveal the predominant underlying neuropathology on its own [48,49]. Inferentially, despite the limited relevant research, perseverations may also manifest in the context of numerous other neurodegenerative entities. Future research ought to focus on the prognostic value of perseverations on the development of other dementia entities (a very small number of non-AD dementia cases were documented in our study, preventing further sub-analyses) as well as other neurodegenerative disorders, for example, Parkinson’s disease. More importantly, the combined value of perseverations with other concomitant cognitive deficits should be explored, since the complete “neuropsychological profile” of an individual might be much more reflective of the exact underlying pattern of neurodegeneration compared to the use of individual cognitive tests.

### Strengths and Limitations

HELIAD is a multidisciplinary, population-based, generational cohort involving a fairly representative sample of the older Greek population. The diagnostic classification of the participants was based on comprehensive neurological and neuropsychological evaluations and was established during expert consensus meetings involving senior neurologists and neuropsychologists, according to standard clinical criteria. All analyses accounted for measures of global cognition and important sociodemographic parameters that may confound language impairment and dementia-MCI development.

Despite the unique qualities of our study, it is important to acknowledge the presence of several limitations. First, the diagnosis of dementia was exclusively based on clinical criteria and did not include imaging or biological biomarkers (potential misclassification bias). Second, the non-trivial dropout rate at follow-up may be responsible for the introduction of non-response bias. Furthermore, despite adjusting our analyses for several crucial parameters, residual confounding should be expected. However, given the small number of total ‘‘events’’ at follow-up, the inclusion of additional covariates in our models could underpower our analyses and conceal non-trivial associations. Moreover, the moderate duration of the follow-up period (approximately 3 years) and the relatively small set of participants with MCI might have underpowered several analyses. Finally, the number of incident dementia cases other than AD was particularly small and did not allow their separate investigation.

## 5. Conclusions

The present study demonstrated that perseverations on the SVF task were associated with an elevated risk of incident all-cause dementia and AD in older, CN individuals. The numbers of clusters, switches, or intrusions were not related to the hazard of developing all-cause dementia or AD in participants with normal cognition. Finally, qualitative VF indices were not linked to the risk of progression to MCI or the hazard of conversion from MCI to all-cause dementia or AD.

## Figures and Tables

**Figure 1 medicina-58-01814-f001:**
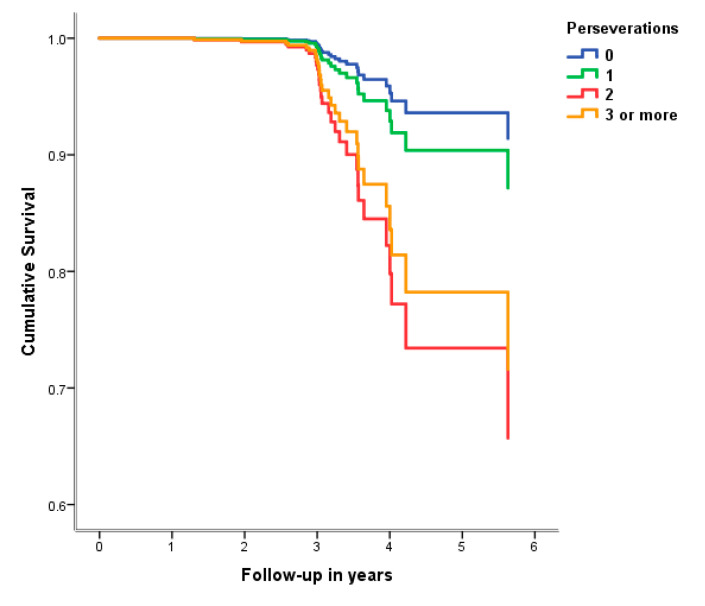
Survival curves for incident dementia according to the perseverations on the semantic task. Owing to their small number, individuals with ≥3 perseverations were clustered together (*n* = 40).

**Table 1 medicina-58-01814-t001:** Baseline characteristics of cognitively normal participants according to the diagnosis of dementia or not at follow-up.

	Baseline Parameter	Without Dementia at Follow-Up (n = 921)	With Dementia at Follow-Up (n = 34)	*p*-Value (between-Group Differences)
	Age in years at baseline (N = 955)	72.75 ± 4.86	77.34 ± 4.99	**<0.001**
	Years of education (N = 955)	8.39 ± 4.86	6.18 ± 4.68	**0.009**
	Sex (M/F) (N = 955)	363/558 (39.4%/60.6%)	15/19 (44.1%/55.9%)	0.582
	Main occupation (Manual/mental) (N = 860)	520/307 (62.9%/37.1%)	24/9 (72.7%/27.3%)	0.250
	Socioeconomic status (Low/high) (N = 955)	395/526 (42.9%/57.1%)	18/16 (52.9%/47.1%)	0.245
	MMSE (N = 937)	27.66 ± 2.16	25.62 ± 2.89	**<0.001**
**Semantic condition**	Number of clusters (N = 936)	2.42 ± 1.31	1.48 ± 1.06	**<0.001**
	Number of switches (N = 936)	10.19 ± 4.73	7.06 ± 4.31	**<0.001**
	Number of intrusions (N = 936)	0.62 ± 1.54	0.35 ± 0.80	0.346
	Number of perseverations (N = 936)	0.57 ± 1.03	0.71 ± 1.01	0.469
**Phonemic** **condition**	Number of clusters (N = 914)	0.34 ± 0.79	0.27 ± 0.52	0.596
	Number of switches (N = 914)	6.91 ± 4.18	4.60 ± 2.88	**0.003**
	Number of intrusions (N = 914)	0.28 ± 0.76	0.23 ± 0.63	0.749
	Number of perseverations (N = 914)	0.41 ± 0.88	0.33 ± 0.84	0.656

n: total number of participants; N: number of participants with available data per variable; M/F: male/female; MMSE: Mini-Mental State Examination; **bold** denotes statistical significance.

**Table 2 medicina-58-01814-t002:** Baseline characteristics of cognitively normal participants based on the diagnosis of mild cognitive impairment (MCI) or not at follow-up.

	Baseline Parameter	Without Dementia or MCI at Follow-Up (n = 761)	With MCI at Follow-Up (n= 160)	*p*-Value (between-Group Differences)
	Age in years at baseline (N = 921)	72.46 ± 4.70	74.13 ± 5.36	**<0.001**
	Years of education (N = 921)	8.74 ± 4.86	6.72 ± 4.52	**<0.001**
	Sex (M/F) (N = 921)	297/464 (39.0%61.0%)	66/94 (41.3%/58.7%)	0.601
	Main occupation (Manual/mental) (N = 827)	414/269 (60.6%/39.4%)	106/38 (73.6%/26.4%)	**0.003**
	Socioeconomic status (Low/high) (N = 921)	313/448 (41.1%/58.9%)	82/78 (51.3%/48.7%)	**0.019**
	MMSE (N = 905)	27.78 ± 2.00	27.09 ± 2.50	**<0.001**
**Semantic condition**	Number of clusters (N = 905)	2.45 ± 1.24	2.27 ± 1.59	0.131
	Number of switches (N = 905)	10.56 ± 4.77	8.41 ± 4.09	**<0.001**
	Number of intrusions (N = 905)	0.63 ± 1.62	0.55 ± 1.09	0.583
	Number of perseverations (N = 905)	0.59 ± 1.01	0.48 ± 1.14	0.202
**Phonemic condition**	Number of clusters (N = 884)	0.35 ± 0.64	0.32 ± 1.31	0.714
	Number of switches (N = 884)	7.07 ± 4.23	6.12 ± 3.80	**0.012**
	Number of intrusions (N = 884)	0.25 ± 0.68	0.42 ± 1.08	**0.011**
	Number of perseverations (N = 884)	0.40 ± 0.84	0.42 ± 1.06	0.781

n: total number of participants; N: number of participants with available data per variable; M/F: male/female; MMSE: Mini-Mental State Examination; **bold** denotes statistical significance.

**Table 3 medicina-58-01814-t003:** Associations between qualitative verbal fluency indices and incident all-cause dementia (primary outcome) and Alzheimer’s dementia (secondary outcome).

Variable	Dementia at Follow-Up	Alzheimer’ s Dementia at Follow-Up
**Semantic condition**	**Adjusted HR (95%CI), *p*-value**	**Adjusted HR (95%CI), *p*-value**
Number of clusters	0.69 (0.49, 0.97), 0.035	0.71 (0.49, 1.03), 0.069
Number of switches	0.87 (0.79, 0.97), 0.012	0.86 (0.77, 0.97), 0.011
Number of intrusions	0.91 (0.60, 1.39), 0.660	0.72 (0.38, 1.34), 0.296
Number of perseverations	**1.52 (1.15, 2.01), 0.003**	**1.55 (1.18, 2.04), 0.002**
**Phonemic condition**	**Adjusted HR (95%CI), *p*-value**	**Adjusted HR (95%CI), *p*-value**
Number of clusters	1.19 (0.73, 1.94), 0.480	1.17 (0.69, 1.98), 0.552
Number of switches	0.88 (0.77, 1.01), 0.060	0.88 (0.76, 1.01), 0.071
Number of intrusions	0.99 (0.55, 1.78), 0.975	1.04 (0.59, 1.84), 0.894
Number of perseverations	1.26 (0.81, 1.96), 0.308	1.32 (0.85, 2.03), 0.217

HR: hazard ratio; CI: confidence interval; **bold** denotes statistical significance.

**Table 4 medicina-58-01814-t004:** Associations between qualitative verbal fluency indices and incident all-cause dementia (primary outcome) and Alzheimer’s dementia (secondary outcome).

Variable	Dementia at Follow-Up	Alzheimer’ s Dementia at Follow-Up
**Semantic condition**	**Adjusted HR (95%CI), *p*-value**	**Adjusted HR (95%CI), *p*-value**
Number of clusters	0.64 (0.43, 0.97), 0.037	0.64 (0.41, 1.01), 0.057
Number of switches	0.94 (0.83, 1.07), 0.364	0.96 (0.84, 1.10), 0.576
Number of intrusions	0.86 (0.56, 1.31), 0.468	0.89 (0.59, 1.34), 0.567
Number of perseverations	1.08 (0.71, 1.66), 0.719	1.01 (0.66, 1.55), 0.950
**Phonemic condition**	**Adjusted HR (95%CI), *p*-value**	**Adjusted HR (95%CI), *p*-value**
Number of clusters	1.20 (0.41, 3.52), 0.742	1.32 (0.42, 4.09), 0.634
Number of switches	1.00 (0.85, 1.16), 0.957	0.98 (0.83, 1.16), 0.844
Number of intrusions	3.05 (0.94, 9.95), 0.062	3.46 (1.07, 11.19), 0.039
Number of perseverations	1.48 (0.47, 4.66), 0.506	1.49 (0.46, 4.80), 0.502

HR: hazard ratio; CI: confidence interval.

**Table 5 medicina-58-01814-t005:** Associations between qualitative verbal fluency indices and incident mild cognitive impairment (MCI).

Variable	MCI at Follow-Up	Variable	MCI at Follow-Up
**Semantic Condition**	**Adjusted HR (95%CI), *p*-Value**	**Phonemic Condition**	**Adjusted HR (95%CI), *p*-Value**
Number of clusters	1.03 (0.90, 1.18), 0.689	Number of clusters	1.21 (0.96, 1.51), 0.101
Number of switches	0.95 (0.91, 0.99), 0.014	Number of switches	0.98 (0.93, 1.04), 0.544
Number of intrusions	1.03 (0.89, 1.18), 0.773	Number of intrusions	1.27 (1.06, 1.53), 0.009
Number of perseverations	1.15 (0.96, 1.39), 0.139	Number of perseverations	1.14 (0.94, 1.38), 0.186

HR: hazard ratio; CI: confidence interval; MCI: mild cognitive impairment.

## Data Availability

The data that support the findings of this study are available from the corresponding author upon reasonable request.

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
