# Peer review of "Qualitative Verbal Fluency Components as Prognostic Factors for Developing Alzheimer’s Dementia and Mild Cognitive Impairment: Results from the Population-Based HELIAD Cohort"

_medicina, 2022, doi:10.3390/medicina58121814_

Round 1

Reviewer 1 Report (Previous Reviewer 2)

Satisfactory

Author Response

Dear Reviewer 1, 

thank you very much for your time and your valuable feedback

Reviewer 2 Report (Previous Reviewer 1)

The first two of my observations have been satisfactorily addressed by the authors. Considering the third observation, a diagnosis without biomarkers reduces the accuracy of the diagnosis since the pathophysiology of neurodegenerative diseases cannot be accurately inferred by a well-structured clinical evaluation. But in this study, the differential diagnosis is not a key point, and just for this reason, the clinical characterization of verbal indices is necessary.

I believe their response to the fourth point is insufficient, and I have some further observations. I brought up the issue that the clinical meaning of the hazard ratio of the indices of the verbal fluencies depends on the comparison with the other tests. In this sense, the list of neuropsychological tests adopted without any descriptive table reporting their values does not address my observation. Considering Table 1, at least the values of the main tests adopted in the study should have been compared with the verbal indices studied. Just as an example, I would like to know the values of Greek verbal learning in the 921 subjects without dementia at follow-up and in the 34 subjects with dementia at follow-up. Also, I would like to know if the hazard ratio of the indices of semantic and verbal fluencies is statistically different from the hazard ratio of the Greek verbal learning test. Moreover, I would like to know the same data for the subjects reported in Table 2.

Round 2

Reviewer 2 Report (Previous Reviewer 1)

The authors responded to the observations although, in my view, the clinical implications remain unclear.

This manuscript is a resubmission of an earlier submission. The following is a list of the peer review reports and author responses from that submission.

Round 1

Reviewer 1 Report

The paper describes an interesting topic related to the potential prognostic value of several indices (clustering, switching, intrusions, and perseverations) obtainable by large used neuropsychological tests as the verbal fluencies of the verbal fluency task (both for semantic and phonemic conditions) in the preclinical stages of mild cognitive impairment (MCI) and dementia.

Even if the aim is intriguing the paper suffers of several criticisms. The authors do not describe the inclusion and exclusion criteria; instead, they mention the reference of these data. A key component of a scientific study is the description of the characteristics of the participants. Hence, the invitation to the reader to search for this data in another paper makes reading obscure.

The evaluation of category and verbal letter fluency, as well as the computation of the number of clusters and the number of switches, are other essential elements that are not given in the same way. In brief, the authors mention core points described in other manuscript instead to describe them in the submitted one. However, I believe that there are two reasons why this is not the best course of writing. First, the paper's exposition in this way becomes unclear. Second, the authors should be open to a new evaluation of their methods even if they have been accepted by other peer-reviewed journals.

Moreover, the diagnostic criteria adopted are out of date. No one biomarker has been used. The description of the neuropsychological battery is limited to the use of MMSE and verbal fluencies (semantic verbal fluency and phonemic verbal fluency). Typically, neuropsychological battery examinations include a number of tests that probe various areas. The authors should have explained why they do not describe other tests as memory tests, executive tests, and so on. The clinical meaning of the hazard ratio of the verbal indices described would have increased compared with other tests, while the hazard ratios reported, related only to the verbal fluencies, have poor clinical meaning.

Reviewer 2 Report

1. Please provide a complete description of every abbreviation at the first presentation. Revise all abbreviations.

2. Grammatical English should be addressed. The authors should simplify the structures of the phrases or request professional English editing. E.g., long phrases and uncommon punctuation.

3. Please, if the authors did the study as STROBE guidelines. They should upload the checklist as supplementary non-published material.

4. Statistics. How were the variables distributed?

5. The reviewer would like to ask the authors, “Why is the data from the HELIAD Cohort not publicly available?” It is an important study of dementia. Also, other authors could access the data and have different hypotheses.
